# FEED-FORWARD LATENT DOMAIN ADAPTATION

## ABSTRACT

We study the highly practical but comparatively under-studied problem of latent-domain adaptation, where a source model should be adapted to a target dataset that contains a mixture of unlabelled domain-relevant and domain-irrelevant examples. Furthermore, motivated by the requirements for data privacy and the need for embedded and resource-constrained devices of all kinds to adapt to local data distributions, we focus on the setting of feed-forward source-free domain adaptation, where adaptation should not require access to the source dataset, and also be back propagation-free. Our solution is to meta-learn a network capable of embedding the mixed-relevance target dataset and dynamically adapting inference for target examples using cross-attention. The resulting framework leads to consistent improvement on strong ERM baselines. We also show that our framework sometimes even improves on the upper bound of domain-supervised adaptation, where only domain-relevant instances are provided for adaptation. This suggests that human annotated domain labels may not always be optimal, and raises the possibility of doing better through automated instance selection.

## 1 INTRODUCTION

Domain shift presents a real-world challenge for the application of machine learning models because performance degrades when deployment data are not from the training data distribution. This issue is ubiquitious as it is often impossible or prohibitively costly to pre-collect and annotate training data that is sufficiently representative of test data statistics. The field of domain adaptation (Kouw & Loog, 2021; Csurka et al., 2022) has therefore attracted a lot of attention with the promise of adapting models during deployment to perform well using only unlabeled deployment data. The main body of work in deep domain adaptation assumes that there is a pre-specified source domain and a pre-specified target domain. An unlabeled adaptation set is provided from the target domain, and various methods define different learning objectives that update a deep model on the unlabeled adaptation set, with the aim of improving performance on new test data drawn from the target domain.

In this paper we make two main contributions: A conceptual contribution, of a highly practical variant of the domain adaptation problem; and an algorithm for effective domain adaptation in this condition.

**A motivating scenario** Let us introduce one potential illustrative application scenario that motivates the variant of the domain adaptation problem that we propose here. Suppose that a robot or other mobile embedded vision system needs to recognise objects. Because it is mobile, it may encounter objects in different unconstrained contexts, e.g., indoor or outdoor backgrounds, sunny or rainy weather, rooms with lights on or lights off, etc. The robot's object recognition model should adapt in order to maintain strong performance across all these conditions, for example by adapting based on a buffer of recently experienced unlabelled images. However, unlike standard pre-defined domain adaptation benchmarks with neatly curated domains, there are two new challenges: 1) Using such a buffer as the adaptation set means that the adaptation data can be of mixed relevance to the test image to be processed at any given instant. For example the recent history used for adaptation may span multiple rooms, while any individual test image comes from a specific room. 2) The adaptation needs to happen on-board the robot and ideally happen in real-time as the adaptation set itself is updated over time. The first challenge is the *latent domain* challenge, wherein uncurated adaptation sets do not have consistent relevance to a given test image (Fig. 1) The second challenge requires adaptation to take place without back-propagation (which is too costly and not supported on most embedded platforms). It means adaptation should be *feed-forward*.

**Latent domain adaptation**    While domain adaptation is very well studied (Kouw & Loog, 2021; Csurka et al., 2022), the vast majority of work assumes that instances have been pre-grouped into one or more subsets (domains) that differ statistically across groups, while being similar within groups. We join a growing minority (Mancini et al., 2021; Deecke et al., 2022; Hoffman et al., 2014; Wang et al., 2022) in arguing that this is an overly restrictive assumption that does not hold in most real applications of interest. Some collection processes may not provide meta-data suitable for defining domain groupings. Alternatively, for other data sources that occur with rich meta-data there may be no obviously correct grouping and existing domain definitions may be sub-optimal (Deecke et al., 2022). Consider the popular iWildCam (Beery et al., 2020) benchmark for animal detection within the WILDS Koh et al. (2021) suite. The default setup within WILDS defines domains by camera ID. But given that images span different weather conditions and day/night cycles as well as cameras, such domains may neither be internally homogenous, nor similarly distinct. There may be more transferability between images from nearby cameras at similar times of day than between images from the same camera taken on a sunny day vs a snowy night. As remarked by Hoffman et al. (2014); Wang et al. (2022), domains may more naturally define a continuum, rather than discrete groups. That continuum may even be multi-dimensional – such as timestamp of image and spatial proximity of cameras. Our latent domain formulation of the domain adaptation problem spans all these situations where domains are hard to define, while aligning with the requirements of real use cases.

**Feed-forward domain adaptation**    Unsupervised domain adaptation aims to adapt models from source datasets (e.g. ImageNet) to the peculiarities of target data distributions in the wild. The mainstream line of work here updates models by backpropagation on an adaptation set from the target data distribution (Kouw & Loog, 2021; Csurka et al., 2022) (and often simultaneously uses the source data (Liang et al., 2020)). We consider adaptation under the practical constraints of an edge device, namely that neither the hardware capability nor the software stack support back-propagation. Therefore we focus on the *feed-forward* condition where adaptation algorithms should use only feed-forward operations, and only the target dataset (source free condition (Liang et al., 2020)). For example, simply updating batch normalisation statistics, which can be done without back-propagation, provides a strong baseline for backprop-free adaptation (Schneider et al., 2020; Zhang et al., 2021).

**Our solution**    To solve the challenge posed above, we propose a feed-forward adaptation framework based on cross-attention between test instances and the adaptation set. The cross-attention module is meta-learned based on a set of training domains, inspired by Zhang et al. (2021). This is a one-off cost paid up-front and performed on a server, after which the actual adaptation is fast. The deployed recognition model flexibly enables each inference operation to draw upon any part of the target adaptation set, exploiting each adaptation instance to a continuous degree. This can improve performance by eliminating adaptation instances that would be conventionally in-domain yet lead to negative transfer (e.g., same camera/opposite time of day), and include transfer from adaptation instances that would conventionally be out-of-domain but could benefit transfer (e.g., similar images/different camera). Our experiments show that our cross-attention approach provides useful adaptation in this highly practical setting across a variety of synthetic and real benchmarks.

## 2   BACKGROUND AND RELATED WORK

**Test-time domain adaptation**    TTDA has emerged as a topical adaptation scenario that focuses on model adaptation without access to source data at adaptation time (i.e., source-free condition), and further adapts to each mini-batch at test time, aligning with an online adaptation scenario. A meta-learning framework for TTDA has been recently proposed under the name adaptive risk minimization (ARM) (Zhang et al., 2021). ARM provides a variety of options for how TTDA is done, including context network that embeds information from the whole minibatch, updates to batch normalization statistics and gradient-based fine-tuning on the minibatch. ARM learns to do TTDA by meta-learning across a large number of tasks. TENT (Wang et al., 2021) is another TTDA method and is based on optimizing channel-wise affine transformation according to the current minibatch.

**Latent domains**    In settings with latent domains, information about domains is not available i.e. there are no domain labels. Further, some domains may be more similar to each other so the boundaries between domains are often blurred. Various approaches have been proposed to deal with latent domains, e.g. sparse latent adapters (Deecke et al., 2022), domain agnostic learning (Peng et al., 2019) that disentangles domain-specific features from class information using an autoencoder

and (Mancini et al., 2021) that discovers multiple latent domains using a specialized architecture with multiple branches. However, these methods focus on standard domain adaptation or supervised learning that is done across many epochs of training.

**Meta-learning** Meta-learning can take various forms (Hospedales et al., 2021), but one of the most common is episodic meta-learning where the learning is done across tasks. As part of episodic meta-learning, the goal is to learn meta-parameters that will help achieve strong performance when presented with a new task. The most popular application of episodic meta-learning is few-shot learning, where we try to e.g. learn to distinguish between different classes after seeing only a few examples of each class. Example meta-parameters include weight initialization (Finn et al., 2017; Antoniou et al., 2019; Li et al., 2017) or metric space as is the case in Prototypical networks (Snell et al., 2017) and RelationNet (Sung et al., 2018). Prototypical networks and RelationNet are both examples of feed-forward methods. Episodic meta-learning has also been used in domain generalization (Li et al., 2019) and as part of the ARM framework (Zhang et al., 2021). We also use the episodic meta-learning paradigm.

**Transformers** As part of our method, we use cross-attention that takes inspiration from the attention mechanism found in the transformer architecture (Vaswani et al., 2017). After transformers became common in natural language processing, they have also led to strong results within computer vision, most prominently as part of the ViT model (Dosovitskiy et al., 2021). ViT model has served as foundation for more recent vision transformers, including CrossViT (Chen et al., 2021) that combines strong performance with efficiency. Our cross-attention mechanism is broadly inspired by the CrossViT cross-attention module (Chen et al., 2021). Our approach has also been inspired by the idea of non-parametric transformers (Kossen et al., 2021) that are able to reason about relationships between data points. Different to CrossViT, we use image-image attention, rather patch-patch, and show how to exploit this for feed-forward source-free latent domain adaptation. Cross-attention has also found use in few-shot classification (Hou et al., 2019; Doersch et al., 2020; Ye et al., 2020), but these approaches use it to obtain better representations of class prototypes for few-shot supervised learning rather than to reason about which unlabelled examples come from relevant domains to drive many-shot unsupervised learning.

## 3 METHODS

### 3.1 SET-UP

**Preliminaries** During deployment the high-level assumption made by many source-free domain adaptation frameworks is that we are provided with a predictive model $f_\psi$ and an unlabeled target adaptation dataset $\boldsymbol{x}_s$ whose label-space is the same as that of the pre-trained model (Liang et al., 2020). Given these, source-free DA approaches define some algorithm $\mathcal{A}$ that ultimately leads to classifying a test instance $x_q$ as $y_q \approx \hat{y}_q = \mathcal{A}(x_q, \boldsymbol{x}_s, \psi)$. There are numerous existing algorithms for this. For example, pseudo-label strategies (Liang et al., 2020; Li et al., 2020; Yang et al., 2021) proceed by estimating labels $\hat{\boldsymbol{y}}_s$ for the adaptation set $\boldsymbol{x}_s$, treating these as ground-truth, backpropagating to update the model $\psi'$ such that it predicts $\hat{\boldsymbol{y}}_s$, and then classifying the test point as $f_{\psi'}(x_q)$. We address the *feed-forward* setting where algorithm $\mathcal{A}$ should not use back-propagation. For example, BN-based approaches (Schneider et al., 2020; Zhang et al., 2021) use the adaptation set $\boldsymbol{x}_s$ to update the BN statistics in $\psi$ as $\psi'$ and then classify the test point as $f_{\psi'}(x_q)$.

While the conventional domain adaptation setting assumes that target domain test examples $x_q$ and target domain training examples $\boldsymbol{x}_s$ are all drawn from a common distribution, the *latent domain* assumption has no such requirement. For example, $\boldsymbol{x}_s$ may be drawn from a mixture distribution and $x_q$ may be drawn from only one component of that mixture. In this case only a subset of elements in $\boldsymbol{x}_s$ may be relevant to adapting the inference for $x_q$.

**Deployment phase** Rather than explicitly updating model parameters, we aim to define a flexible inference routine $f_\psi$ that processes both $x_q$ and $\boldsymbol{x}_s$ to produce $\hat{y}_q$ in a feed-forward manner, i.e., $\hat{y}_q = \mathcal{A}(x_q, \boldsymbol{x}_s, \psi) = f_\psi(x_q, \boldsymbol{x}_s)$. In this regard our inference procedure follows a similar flow to variants of ARM (Zhang et al., 2021), with the following key differences: (1) ARM is transductive: it processes a batch of instances at once without distinguishing test instances and target adaptation set, so elements $x_q$ are members of $\boldsymbol{x}_s$. (2) ARM makes the conventional domain-observed assumption

Figure 1: Illustration of the standard and latent domain adaptation problem settings. In the LDA setting (support) images come from a variety of domains of mixed and unknown relevance to the test (query) image. In standard DA adaptation images are all assumed to be equally relevant.

that domains have been defined by an external oracle that ensures all $x_q$ and $\boldsymbol{x}_s$ are drawn from the same distribution. We do not make this assumption and assess robustness to irrelevant elements in $\boldsymbol{x}_s$.

**Training phase**  To train a model that can be used as described above, we follow an episodic meta-learning paradigm (Zhang et al., 2021; Hospedales et al., 2021). This refers to training $f_\psi$ using a set of simulated domain adaptation tasks. At each iteration, we generate a task with a unique pair of query and support instances $(\boldsymbol{x}_s, (y_q, x_q))$ keeping label space the same across all tasks. We simulate training episodes where $\boldsymbol{x}_s$ contains instances with varying relevance to $x_q$. The goal is for $f_\psi$ to learn how to select and exploit instances from $\boldsymbol{x}_s$ in order to adapt inference for $x_q$ to better predict $y_q$.

In particular, our task sampler defines each task as having support examples uniformly sampled across a random set of $N_D$ domains, with the query example being from one of these domains. More formally, each task can be defined as:

$$\mathcal{T} = \left\{ \left\{ x_{s,1}, x_{s,2}, \ldots x_{s,N_s} \right\}, x_q, y_q \right\}$$

for $N_s$ unlabelled support examples $x_{s,\cdot}$ and query example $x_q$ with label $y_q$.

**Example**  We give an illustration of a task in Figure 1. The chosen example comes from the real-world iWildCam dataset (Beery et al., 2020), where cameras at different locations are meant to act as different domains. This is challenging, as our model is not provided with this domain/camera annotation in the latent DA setting, and must learn to estimate relevance. On the other hand, we can see from this example that sometimes images from the same annotated domain (= camera) are not very similar, and conversely images from other domains may be quite similar. It may be possible for a model in this setup to do better than standard observed-domain methods that assume an external process provides exclusively same-domain data for adaptation. Our experiments will confirm that this can sometimes happen.

## 3.2 OBJECTIVE

Our goal is to train a model that can adapt to relevant examples from the support set and obtain superior performance on the query examples. We can formalize this using the following objective:

$$\min_{\boldsymbol{\theta}, \boldsymbol{\phi}, \boldsymbol{\omega}} \mathcal{E}(\boldsymbol{\theta}, \boldsymbol{\phi}, \boldsymbol{\omega}) = \mathbb{E}_{p_d|\{d_i\}} \left[ \mathbb{E}_{p_{\{x_q, y_q|d\}}, \{x_s|\{d_i\}\}} \left[ \frac{1}{N_q} \sum_{k=1}^{N_q} \ell(f_{\boldsymbol{\phi}}(f_{\boldsymbol{\theta} \circ \boldsymbol{\omega}}(x_{q,k}; \boldsymbol{x}_s), y_{q,k})) \right] \right], \quad (1)$$

where $\psi = \{\boldsymbol{\theta}, \boldsymbol{\phi}, \boldsymbol{\omega}\}$ are the parameters of the feature extractor, classifier and cross-attention module respectively (described in detail next), $\boldsymbol{x}_s$ are the support examples used for adaptation, while $\boldsymbol{x}_q$ are the query examples for which we make predictions and come from domain $d$. The support examples come from a set of domains $\{d_i\}$ with $d \in \{d_i\}$. There are $N_q$ query examples and $\mathcal{E}$ represents the generalization error after adapting the model.

### 3.3 ARCHITECTURE

The key to solving Eq. 1 is defining an architecture $f_\psi$ that can identify and exploit relevant support instances within $\boldsymbol{x}_s$. Our solution to this relies on cross-attention between query and support images. We first embed the support and query examples using the same feature extractor, after which we pass the embeddings through the cross-attention module. The cross-attention module gives us transformed query examples that are then added to the embeddings of the query examples as a residual connection, after which the classifier makes predictions. Compared to CrossViT (Chen et al., 2021), we do cross-attention between support and query images from different domains, image-to-image rather than patch-to-patch and on extracted features right before the classifier layer.

**Cross-attention module**    Given a set of support examples $\boldsymbol{x}_s$ and query examples $\boldsymbol{x}_q$, we use the feature extractor $f_\theta$ to extract features $f_\theta(\boldsymbol{x}_s)$, $f_\theta(\boldsymbol{x}_q)$. Cross-attention module $\mathtt{CA}_{\boldsymbol{\omega}}(f_\theta(\boldsymbol{x}_s); f_\theta(\boldsymbol{x}_q))$ parameterized by $\boldsymbol{\omega}$ then transforms query embeddings $f_\theta(\boldsymbol{x}_q)$, using support embeddings $f_\theta(\boldsymbol{x}_s)$ as keys. The output of the cross-attention module is added to the query example features as a residual connection, which is then used by the classifier $f_\phi$ to predict labels of the query examples $\hat{\boldsymbol{y}}_q = f_\phi(f_\theta(\boldsymbol{x}_q) + \mathtt{CA}_{\boldsymbol{\omega}}(f_\theta(\boldsymbol{x}_s); f_\theta(\boldsymbol{x}_q)))$.

The cross-attention module itself performs image-to-image cross-attention, rather than patch-to-patch. More specifically, after extracting the features we flatten all spatial dimensions and channels into one vector, which represents the whole image. Image-to-image attention is more suitable for domain adaptation than patch-based option because the overall representation should better capture the nature of the domain rather than a patch. Another benefit of image-to-image attention is also that it is significantly more efficient – we attend to the whole image rather than patches, which makes the overall computations manageable even with more images.

Our cross-attention module is parameterized by a set of learnable projection matrices $\boldsymbol{W}_q, \boldsymbol{W}_k, \boldsymbol{W}_v$ (all of size $\mathbb{R}^{C \times (C/R)}$) with additional projection matrix $\boldsymbol{W} \in \mathbb{R}^{(C/R) \times C}$ to transform the queried outputs (we refer to all of these parameters collectively as $\boldsymbol{\omega}$). The output of the feature extractor $f_\theta$ is flattened into one vector (any spatial information is flattened), giving $C$ channels, so $f_\theta(\boldsymbol{x}_q) \in \mathbb{R}^{N_q \times C}, f_\theta(\boldsymbol{x}_s) \in \mathbb{R}^{N_s \times C}$. We also specify ratio $R$ that allows us to use rectangular projection matrices with fewer parameters, which improves efficiency and also provides regularization.

Formally we express $\mathtt{CA}_{\boldsymbol{\omega}}$ as:

$$\boldsymbol{q} = f_\theta(\boldsymbol{x}_q)\boldsymbol{W}_q, \quad \boldsymbol{k} = f_\theta(\boldsymbol{x}_s)\boldsymbol{W}_k, \quad \boldsymbol{v} = f_\theta(\boldsymbol{x}_s)\boldsymbol{W}_v,$$

$$\boldsymbol{A} = \mathtt{softmax}\left(\boldsymbol{q}\boldsymbol{k}^T / \sqrt{C/h}\right), \quad \mathtt{CA}_{\boldsymbol{\omega}}(f_\theta(\boldsymbol{x}_s)) = \boldsymbol{A}\boldsymbol{v}.$$

Similarly as CrossViT (Chen et al., 2021) and self-attention more broadly, we use multiple heads $h$, so we refer to it as $\mathtt{MCA}$. We also use layer normalization as is the common practice. The output of $\mathtt{MCA}$ is added to the query example embeddings as a residual connection:

$$\boldsymbol{z} = f_\theta(\boldsymbol{x}_q) + \mathtt{MCA}_{\boldsymbol{\omega}}(\mathtt{LN}(f_\theta(\boldsymbol{x}_s)); \mathtt{LN}(f_\theta(\boldsymbol{x}_q))),$$

which is then passed through the classifier $f_\phi$ to obtain predictions $\hat{y} = f_\phi(\boldsymbol{z})$. Following CrossViT, we do not apply a feed-forward network after cross-attention. We directly add the output via residual connection and pass it to the classifier.

### 3.4 META-LEARNING

We train the main model (composed of the feature extractor $f_\theta$ and classifier $f_\phi$ and the cross-attention module (parameterized by $\boldsymbol{\omega}$) by meta-learning across many tasks. Each task has the structure described earlier in Section 3.1. Meta-learning is computationally efficient in this case because the inner loop does not include back-propagation based optimization – the adaptation to the support examples is done purely feed-forward. We provide a summary of both how meta-training and inference (meta-testing) is done in Algorithm 1.

---

**Algorithm 1** Episodic meta-learning for source-free latent domain adaptation with CXDA

---

```
// Meta-training
```

**Require:** # training steps $T$, # latent domains in a task $N_D$, # support examples $N_s$, # query examples $N_q$, learning rate $\eta$

1: **Initialize:** $\boldsymbol{\theta}, \boldsymbol{\phi}, \boldsymbol{\omega}$
2: **for** $t = 1, \ldots, T$ **do**
3:      Sample $N_D$ support domains $\{\mathbb{D}_s\}_1^{N_D}$ from training domains
4:      Sample query domain $\mathbb{D}_q$ from the set of support domains $\{\mathbb{D}_s\}_1^{N_D}$
5:      Sample $N_s$ unlabelled support images $\boldsymbol{x}_s$ uniformly from the selected support domains $\{\mathbb{D}_s\}_1^{N_D}$
6:      Sample $N_q$ labelled query images $\boldsymbol{x}_q, \boldsymbol{y}_q$ from domain $\mathbb{D}_q$
7:      Predict query labels $\hat{\boldsymbol{y}}_q \leftarrow f_{\boldsymbol{\phi}}(f_{\boldsymbol{\theta}}(\boldsymbol{x}_q) + \texttt{MCA}_{\boldsymbol{\omega}}(\texttt{LN}(f_{\boldsymbol{\theta}}(\boldsymbol{x}_s)); \texttt{LN}(f_{\boldsymbol{\theta}}(\boldsymbol{x}_q)))))$
8:      $(\boldsymbol{\theta}, \boldsymbol{\phi}, \boldsymbol{\omega}) \leftarrow (\boldsymbol{\theta}, \boldsymbol{\phi}, \boldsymbol{\omega}) - \eta \nabla_{(\boldsymbol{\theta}, \boldsymbol{\phi}, \boldsymbol{\omega})} \sum_{k=1}^{N_q} \ell(\hat{y}_{q,k}, y_{q,k})$
9: **end for**

```
// Inference on a new task
```

**Require:** $\boldsymbol{\theta}, \boldsymbol{\phi}, \boldsymbol{\omega}$, support $\boldsymbol{x}_s$ and query $\boldsymbol{x}_q$ examples from new domains
10: $\hat{\boldsymbol{y}}_q \leftarrow f_{\boldsymbol{\phi}}(f_{\boldsymbol{\theta}}(\boldsymbol{x}_q) + \texttt{MCA}_{\boldsymbol{\omega}}(\texttt{LN}(f_{\boldsymbol{\theta}}(\boldsymbol{x}_s)); \texttt{LN}(f_{\boldsymbol{\theta}}(\boldsymbol{x}_q)), \boldsymbol{\omega})))$

---

## 4 EXPERIMENTS

### 4.1 BENCHMARKS

We evaluate our approach on a variety of synthetic and real-world benchmarks, namely FEMNIST (Caldas et al., 2018), CIFAR-C (Hendrycks & Dietterich, 2019), TinyImageNet-C (Hendrycks & Dietterich, 2019) and iWildCam (Beery et al., 2020). All of these benchmarks have a large number of domains, e.g. around 100 for CIFAR-C and TinyImageNet-C and around 300 for FEMNIST and iWildCam. Using a large number of domains for pre-training is reasonable as for many practical problems it is possible to collect such pre-training data. We describe each benchmark next.

**FEMNIST** dataset includes images of handwritten letters and digits, and is derived from the EMNIST dataset (Cohen et al., 2017) by treating each writer as a domain. **CIFAR-C** extends CIFAR-10 (Krizhevsky, 2009) by a applying a variety of corruptions such as different brightness, snow or various types of blurring. There are different levels of severity with which the corruptions are applied, giving rise to multiple domains for the different levels. **TinyImageNet-C** is an extension of TinyImageNet analogous to CIFAR-C. **iWildCam** is a large-scale real-world dataset that includes images of different animal species taken by cameras in different locations. There is a lot of variability in the style of images in different cameras, for example different illumination, camera angle or vegetation. The dataset has also substantial class imbalance, so macro F1 score is used for evaluation.

For FEMNIST, CIFAR-C and TinyImageNet-C we follow the splits into meta-training, meta-validation and meta-testing sets as selected in (Zhang et al., 2021). For iWildCam we follow the splits of domains selected in (Koh et al., 2021). Additionally for iWildCam we filter out all domains that have fewer than 40 examples.

### 4.2 BASELINES

**ERM** Empirical risk minimization or ERM is a baseline that simply trains on all training domains and performs no domain adaptation. It is known to work surprisingly well and is often difficult to beat when properly tuned (Gulrajani & Lopez-Paz, 2020). In our case it is trained following the episodic pipeline for fair comparison i.e. it is directly trained using the query examples during meta-training.

**BN** A simple and often useful method for source-free domain adaptation is to update the batch normalization statistics using the unlabelled target domain data Schneider et al. (2020). It has achieved strong results in conventional SFDA (Ishii & Sugiyama, 2021). However, in the latent DA setting it is unclear if statistics calculated across a support set of varying relevance will be helpful for achieving better performance. During evaluation, the statistics are updated using all support examples, and directly used for the query examples.

**CML** Contextual Meta-Learning is the main instantiation of ARM (Zhang et al., 2021) as a way to extract information from the whole minibatch in test-time adaptation and use it to obtain better performance on test images. We apply the CML on the whole support set with images from different domains and then use it as additional information for making predictions on test images. CML is a feed-forward domain adaptation method, but it has not been designed for the latent domain adaptation problem.

**Backprop-based** Fine-tuning with standard domain adaptive losses such as entropy minimisation (Grandvalet & Bengio, 2004) (FT-EM) and infomax (Shi & Sha, 2012) (FT-IM). Fine-tuning with these objectives is widely used in prior SFDA methods, and these baselines roughly correspond to applying methods such as TENT (Wang et al., 2021) and SHOT (Liang et al., 2020) to our problem setting respectively. These results are presented for context rather than fair comparison, because they use backprop (which we do not) and they are not designed for latent domain adaptation (unlike us).

## 4.3 IMPLEMENTATION DETAILS

**Our solution – CXDA** Our cross-attention module first flattens all spatial information and channels into one vector for each image, so it works image-to-image. In line with existing literature (Vaswani et al., 2017; Chen et al., 2021) we use 8 heads and layer normalization on the flattened features of support and query images. The use of layer normalization means that our approach does not rely on a minibatch of query examples i.e. it natively supports streaming mode and does not need mutiple query examples to obtain strong results, unlike existing test-time domain adaptation approaches (Zhang et al., 2021; Wang et al., 2021).

Support images are projected into keys and values, while query images act as queries for cross-attention after transformation by a projection matrix. After calculating the attention map and applying it to the values, we multiply the output by a further projection matrix. We use only one cross-attention layer and our projection matrices have rectangular shape of $C \times C/2$ where $C$ is the dimensionality of the flattened features. No dropout is used.

**Data augmentation** We use weak data augmentation during meta-training. The exact augmentations are cropping, horizontal flipping, small rotations (up to 30 degrees) and are different from the corruptions tested in some of the benchmarks. These are applied with probability 0.5 independently.

**Task sampling** Our tasks have 5 support domains, with 20 examples in each, overall 100 support examples. Query examples come from one randomly selected support set domain (out of 5 options) and there are 20 of them. Note that the method fully supports streaming mode, so no statistics are calculated across the batch and it works independently for each. The exact number of tasks for meta-validation and meta-testing is respectively (420, 420) for FEMNIST, (850, 11000) for CIFAR-C, (1700, 11000) for TinyImageNet-C, and (745, 2125) for iWildCam.

**Training** We follow the hyperparameters used by Zhang et al. (2021) for FEMNIST, CIFAR-C and TinyImageNet-C, and we also train the cross-attention parameters with the same optimizer. For FEMNIST and CIFAR-C a small CNN model is used, while for TinyImageNet-C a pre-trained ResNet-50 (He et al., 2015) is fine-tuned. For iWildCam we follow the hyperparameters selected in (Koh et al., 2021), but with images resized to $112 \times 112$, training for 50 epochs and with mini-batch size resulting from our task design (100 support and 20 query examples). All our experiments are repeated across three random seeds.

**Evaluation metrics** We follow Zhang et al. (2021) in reporting average and worst performance over all testing tasks. While Zhang et al. (2021) reports the worst single task, we modify this metric to report the average performance of the worst decile of tasks. The reason is that for some benchmarks, among all 10,000+ test tasks with varying domain transfer difficulty there can easily be at least one task with zero accuracy, making it a less meaningful measure.

## 4.4 RESULTS

We report our results in Table 1 for all bencharks: FEMNIST, CIFAR-C, TinyImageNet-C and large-scale real-world iWildCam benchmark. We include both average performance as well as reliability via the worst case performance (Zhang et al., 2021), with our bottom decile modification. From the results we can see our cross-attention approach results in consistent improvements over the strong

ERM baseline across all benchmarks, as well as the CML and BN baselines. The encouraging result on iWildCam highlights our method works well also in practical real-world scenarios.

Overall we see CML and BN strategies that naively combine information from all support examples have limited success when the support set has both domain relevant and domain irrelevant examples. In fact, CML achieves lower performance than ERM in some of the benchmarks, despite having a mechanism for domain adaptation. The results show the need to adaptively select the right examples from the support set when they come from domains of mixed relevance. The results confirm our cross-attention based framework can successfully select useful information from the set of examples with both relevant and irrelevant examples and ultimately achieve superior performance. Backprop-based alternatives usually perform worse, despite being slower, due to lack of support for latent domains.

Table 1: Main benchmark resuls: average and worst-case (worst 10% tasks) test performance, with standard error of the mean across 3 random seeds. Accuracy is reported for all except iWildCam, where F1 score is used (%). Feed-forward approaches are at the top and backprop ones at the bottom.

| | FEMNIST | | CIFAR-C | | TinyImageNet-C | | iWildCam | |
|---|---|---|---|---|---|---|---|---|
| Approach | W10% | Avg | W10% | Avg | W10% | Avg | W10% | Avg |
| ERM | $52.7 \pm 1.4$ | $77.2 \pm 0.9$ | $44.3 \pm 0.5$ | $68.6 \pm 0.3$ | $4.8 \pm 0.2$ | $26.4 \pm 0.4$ | $0.0 \pm 0.0$ | $38.7 \pm 0.8$ |
| CML Zhang et al. (2021) | $50.4 \pm 1.3$ | $76.0 \pm 0.9$ | $44.8 \pm 0.5$ | $69.5 \pm 0.5$ | $4.8 \pm 0.5$ | $25.7 \pm 0.6$ | $0.0 \pm 0.0$ | $38.7 \pm 1.1$ |
| BN Ishii & Sugiyama (2021); Zhang et al. (2021) | $52.2 \pm 1.5$ | $78.0 \pm 0.7$ | $45.4 \pm 0.7$ | $69.3 \pm 0.4$ | $5.9 \pm 0.2$ | $27.7 \pm 0.3$ | $1.9 \pm 1.1$ | $42.5 \pm 0.8$ |
| Our CXDA | $53.3 \pm 0.6$ | $78.3 \pm 0.0$ | $49.4 \pm 0.6$ | $72.0 \pm 0.3$ | $6.5 \pm 0.2$ | $28.6 \pm 0.3$ | $3.6 \pm 1.5$ | $43.5 \pm 1.5$ |
| FT-EM Grandvalet & Bengio (2004) | $51.7 \pm 1.4$ | $77.6 \pm 0.8$ | $44.9 \pm 0.6$ | $69.2 \pm 0.4$ | $3.9 \pm 0.4$ | $25.7 \pm 0.3$ | $0.0 \pm 0.0$ | $38.6 \pm 0.8$ |
| FT-IM Shi & Sha (2012); Liang et al. (2020) | $52.5 \pm 1.2$ | $77.5 \pm 0.8$ | $45.6 \pm 0.5$ | $69.5 \pm 0.3$ | $4.8 \pm 0.4$ | $24.6 \pm 1.0$ | $0.0 \pm 0.0$ | $38.7 \pm 0.8$ |

## 4.5 FURTHER ANALYSIS

As part of analysis we study several questions: 1) How does the performance of unsupervised cross-attention compare with a supervised version? 2) How does the inference and training time compare for our cross-attention method and the baselines? 3) What do the attention weights look like and what do they imply? 4) How does the performance vary with variable number of domains in the support sets? 5) How does the size of the support set influence the performance?

**Domain-supervised vs domain-unsupervised adaptation** Recall that our main CXDA algorithm and experiments above are domain unsupervised. This may induce a cost due to distraction by domain-irrelevant adaptation data (e.g., as observed by CML underperforming ERM previously) or a potential benefit due to enabling transfer. We therefore compare our unsupervised method with a domain-supervised alternative, with manually defined attention weights based on domain labels. Table 2 shows the results are dataset dependent. The fact that in at least some cases domain-unsupervised adaptation outperforms the supervised case shows that the benefit can sometimes outweigh the cost, and that it is possible for a suitable model to outperform manual domain annotations.

Table 2: Comparison of unsupervised and supervised CXDA on our benchmarks. Average test accuracy for all benchmarks apart from iWildCam where F1 score is reported (%).

| Cross-attention | FEMNIST | CIFAR-C | TinyImageNet-C | iWildCam |
|---|---|---|---|---|
| Unsupervised | $78.3 \pm 0.0$ | $72.0 \pm 0.3$ | $28.6 \pm 0.3$ | $43.5 \pm 1.5$ |
| Supervised | $79.4 \pm 0.4$ | $69.8 \pm 0.4$ | $28.6 \pm 0.2$ | $52.0 \pm 1.2$ |

**Comparison of inference and training times** We show our approach is fast and can do adaptation quickly in Table 3 (left), with inference time very similar to feed-forward baselines. Table 3 (right) shows that meta-training is longer for the smaller datasets, but the difference is small for large datasets and models. Backpropagation-based approaches have significantly larges inference times than the feedforward ones, showing the need for specialized feedforward adaptation methods. The total runtime is closer to the feedforward approaches because the training is done without any fine-tuning. All experiments within the same benchmark used the same GPU and number of CPUs.

**Analysis of attention weights** We have analysed the attention weights to understand the learned behaviour of the cross-attention mechanism. We have selected the large-scale iWildCam benchmark

Table 3: Comparative computational cost of different adaptation methods for adaptation and training.

| Approach | Inference/Adaptation Time (ms/task) | | | | Total Runtime Train+Adaptation (minutes) | | | |
|---|---|---|---|---|---|---|---|---|
| | FEMNIST | CIFAR-C | TinyImageNet-C | iWildCam | FEMNIST | CIFAR-C | TinyImageNet-C | iWildCam |
| ERM | $12.0 \pm 0.1$ | $15.7 \pm 0.3$ | $52.8 \pm 0.2$ | $352.0 \pm 2.8$ | $38.3 \pm 0.3$ | $26.6 \pm 0.2$ | $94.1 \pm 0.2$ | $440.2 \pm 5.4$ |
| CML | $13.1 \pm 0.2$ | $15.8 \pm 0.1$ | $48.9 \pm 0.1$ | $385.2 \pm 8.0$ | $55.2 \pm 1.0$ | $35.2 \pm 0.2$ | $131.8 \pm 0.2$ | $461.9 \pm 7.2$ |
| BN | $16.5 \pm 0.7$ | $19.2 \pm 0.2$ | $74.1 \pm 0.1$ | $345.2 \pm 1.9$ | $44.5 \pm 0.3$ | $31.6 \pm 0.3$ | $112.3 \pm 0.3$ | $432.2 \pm 0.8$ |
| CXDA | $17.8 \pm 1.5$ | $20.5 \pm 0.2$ | $77.9 \pm 2.8$ | $392.5 \pm 1.3$ | $110.0 \pm 12.0$ | $63.2 \pm 0.6$ | $167.9 \pm 1.9$ | $491.6 \pm 1.0$ |
| FT-EM | $352.7 \pm 42.4$ | $387.1 \pm 10.5$ | $840.2 \pm 15.4$ | $2044.7 \pm 238.0$ | $97.5 \pm 6.4$ | $156.6 \pm 3.6$ | $483.4 \pm 4.9$ | $907.1 \pm 28.0$ |
| FT-IM | $296.9 \pm 3.1$ | $385.9 \pm 10.9$ | $830.7 \pm 16.6$ | $1709.8 \pm 16.5$ | $78.2 \pm 0.4$ | $150.2 \pm 3.9$ | $470.8 \pm 7.3$ | $688.0 \pm 14.8$ |

and used one of the trained cross-attention models. Figure 2 shows the density histogram of attention weights for same and different domain support examples, relative to the query examples in each task. From the plot we observe that: 1) There is a significant amount of weight spent on attending to examples in different domains to the current query. This suggests that the model is exploiting knowledge transfer beyond the boundaries of the standard (camera-wise) domain annotation in the benchmark, as illustrated in Figure 1. 2) Nevertheless overall the weight distribution tends to attend somewhat more strongly to the in-domain instances than out-of-domain instances. This shows that our cross-attention module has successfully learned how to match query instances with corresponding domain instances in the support set, despite never experiencing domain-supervision.

**Variable number of domains** Tables 4 and 5 in Appendix D show that the best performance is obtained when there are fewer domains, confirming our intuition. However, CXDA can handle well also cases when there is a large number of domains and consistently outperforms other approaches irrespective of the number of domains.

**Variable support set size** Tables 6 and 7 in Appendix D analyse the impact of variable number of examples in the support set. The results confirm CXDA is scalable and outperforms other approaches even if the support set size changes to more or fewer examples.

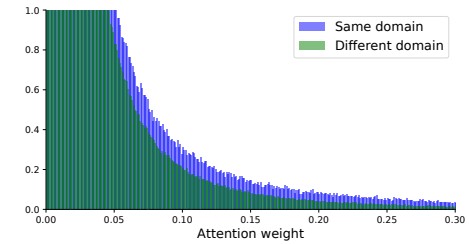

Figure 2: Density histograms of attention weights for pairs of same and different domain examples in the test tasks of iWildCam benchmark.

### 4.6 DISCUSSION

**Scalability and applicability** Our cross-attention approach is fast in the scenario when there are a moderate number of examples for adaptation (in our experiments we use 100), and not too many inferences to be made per adaptation. In this regime it is much faster than the mainstream line of backpropagation-based adaptation solutions (Kouw & Loog, 2021; Csurka et al., 2022; Liang et al., 2020; Wang et al., 2021). Whereas, if there are very many inferences to be made then the overhead cost of back-propagation would eventually be amortized. As for attention more broadly, its computational cost depends on the number of query and support examples, and the approach would be expensive if there were many thousands or millions of examples for both. For a very large support set, one could simply take a random or semi-random subset of images for adaptation. However, the most effective use case is one where the adaptation set is smaller and/or changes rapidly compared to the frequency of inference.

## 5 CONCLUSION

We have introduced a new highly practical setting where we adapt a model using examples that come from a mixture of domains and are without domain or class labels. To answer this new highly challenging adaptation problem, we have developed a novel solution based on cross-attention that is able to automatically select relevant examples and use them for adaptation on the fly.

REPRODUCIBILITY STATEMENT

We include the code for our approach as part of the supplementary material. We also include details for our experiments within the main text or the appendix.

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

# A    QUALITATIVE ANALYSIS

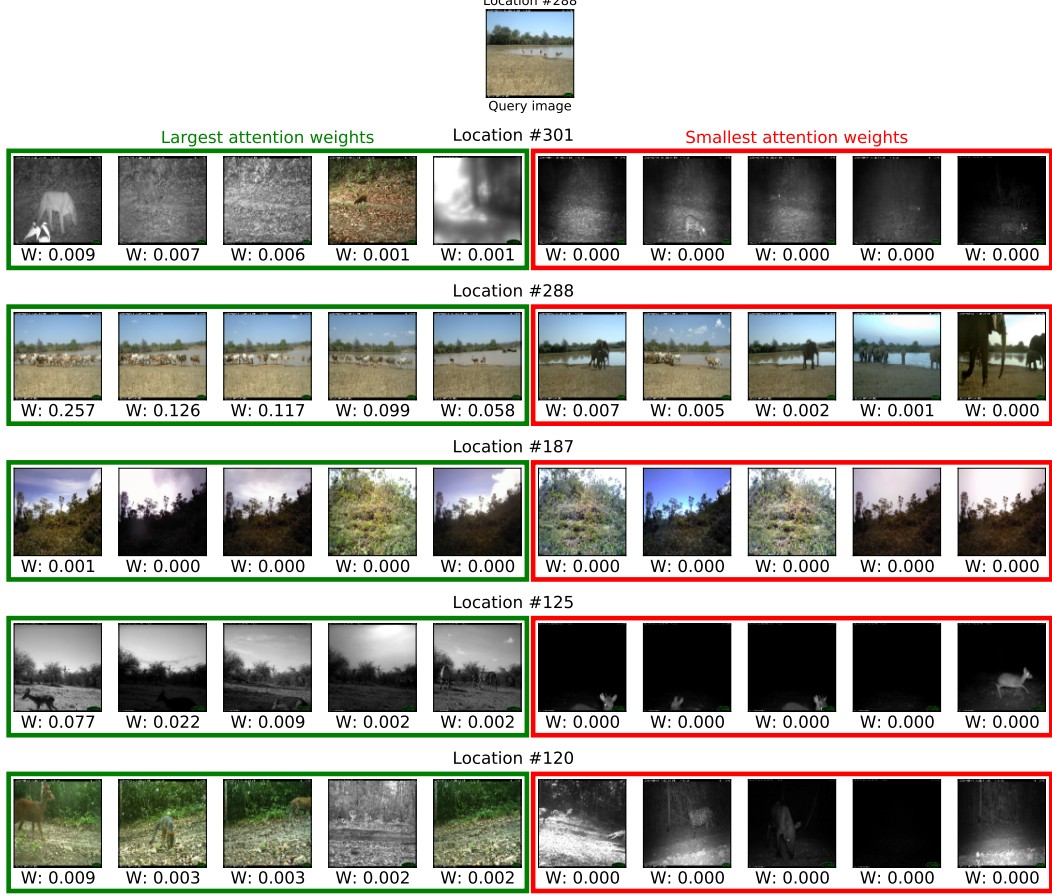

Figure 3: Analysis of attention weights for an example task in iWildCam, with a query image coming from location (camera trap) #288. We show the five support examples in each domain that have the largest and smallest attention weights. Similar images from the same location (#288) are given the largest weights, but also relevant images from other locations (e.g. #125) are given larger weights. The examples with the smallest attention weights visually do not seem relevant.

# B    FORMAL DEFINITION OF LATENT DOMAINS

We provide both high-level and more detailed formal definition of how we use latent domains.

**High-level explanation of latent domains:**    $p(D)$ is a distribution over domains, and $p_d(x, y)$ is the distribution over data in domain $d$. $I$ is a set of domain indexes drawn from $p(D)$, $\mathcal{D}$ is a dataset drawn from several domains within $p(D)$ and the dataset does not contain the domain IDs corresponding to each instance $(x, y)$. Equation 2 samples the domains to be used in a given dataset. Eq. 3 samples the dataset from those domains, but doesn't record the domains.

$$I = \{d : d \sim p(D)\} \tag{2}$$
$$\mathcal{D} = \{(x, y) : (x, y) \sim p_d(x, y), d \in I\} \tag{3}$$

**More detailed explanation of latent domains assumption in each step of our algorithm:**    Eq. 4 says the adaptation (support) and test (query) set are from disjoint domains to the pre-train set. Eq. 6 says the query/test domain should be observed in the support set, but the support set can contain irrelevant domains to the query/test set.

$$I^{pretrain} = \{d : d \sim p(D)\} \tag{4}$$

$$\mathcal{D}^{pretrain} = \{(x, y) : (x, y) \sim p_d(x, y), d \in I^{pretrain}\} \tag{5}$$

$$I^{deploy} = \{d : d \sim p(D), d \notin I^{pretrain}\} \tag{6}$$

$$\mathcal{D}^{support} = \{(x, y) : (x, y) \sim p_d(x, y), d \in I^{deploy}\} \tag{7}$$

$$I^{query} \subseteq I^{deploy} \tag{8}$$

$$\mathcal{D}^{query} = \{(x, y) : (x, y) \sim p_d(x, y), d \in I^{query}\} \tag{9}$$

$$\tag{10}$$

Note that $d$ doesn't have to be either discrete or scalar in our framework, unlike most others.

## C  ADDITIONAL DETAILS

We have followed the experimental choices from Zhang et al. (2021) for FEMNIST, CIFAR-10-C, TinyImageNet-C and Koh et al. (2021) for iWildCam, unless we specify otherwise. This includes the splits of data into training, validation and test sets as well as the splits of domains into meta-training, meta-validation and meta-test sets of domains. For iWildCam we use the OOD splits from Koh et al. (2021).

### C.1  MODELS

Feature extractor and classifier:

- FEMNIST: CNN with three convolutional layers, hidden dimension of 128, batch normalization, ReLU activation, kernel of 5, padding of 2. Classifier consists of two fully-connected layers with 200 hidden units and ReLU activation in between. The input shape of images is $28 \times 28$.

- CIFAR-C: Same architecture as for FEMNIST, but with three input channels instead of 1 (colour images). The input shape of images is $32 \times 32$.

- TinyImageNet-C: ImageNet pre-trained ResNet50 (He et al., 2015). The classifier consists of one fully connected layer. The input shape of images is $64 \times 64$.

- iWildCam: ImageNet pre-trained ResNet50 (He et al., 2015). The classifier consists of one fully connected layer. The input shape of images is $112 \times 112$.

Adaptation-specific components:

- CML: Context network is used to transform the support examples – three convolutional layers, 64 hidden units, kernel size of 5, padding of 2, with batch normalization and ReLU activation. The output of the network has the same shape as input. To create the context we average the context network outputs across the support examples and use the same context for all query examples in the task.

- CXDA: All key details are explained in the main text, and we provide more detailed explanations here. The size of the fully-connected layers depends on the flattened shape of the features – only one cross-attention layer is used. After multiplying attention weights with projected values ($\boldsymbol{Av}$), we transform the output further using projection matrix $\boldsymbol{W}$, similarly as (Chen et al., 2021). However, we do not use a further MLP model that would contain multiple layers and non-linearity, so we also follow (Chen et al., 2021) in this aspect. The output of the cross-attention module has the same shape as input. As part of CXDA, batch normalization statistics of the feature extractor are updated too using the support set.

### C.2  TRAINING

Dataset-specific training details:

- FEMNIST: SGD with learning rate of $10^{-4}$, momentum of 0.9 and weight decay of $10^{-4}$, trained for 200 epochs, with validation set evaluated every 10 epochs, and early stopping based on accuracy.
- CIFAR-C: SGD with learning rate of $10^{-2}$, momentum of 0.9 and weight decay of $10^{-4}$, trained for 100 epochs, with validation set evaluated every 10 epochs, and early stopping based on accuracy.
- TinyImageNet-C: SGD with learning rate of $10^{-2}$, momentum of 0.9 and weight decay of $10^{-4}$, trained for 50 epochs, with validation set evaluated every 5 epochs, and early stopping based on accuracy.
- iWildCam: Adam with learning rate of $3 \times 10^{-5}$, no weight decay, trained for 50 epochs, with validation set evaluated every 5 epochs, and early stopping based on macro F1 score.

In all cases we use cross-entropy loss, and the cross-attention parameters are optimized in the same way as the main model. In each iteration we use a task that has 5 domains with 20 support examples for each sampled domain, and there are 20 query examples from one selected domain from the set of current domains.

Details about fine-tuning (FT) of the pre-trained ERM model: we perform fine-tuning by taking 10x smaller learning rate compared to the ones used during training and then performing 10 steps on the task's support data. We consider two losses: 1) entropy minimization (EM) and 2) information maximization (IM) of the support set example predictions during the fine-tuning. We reset the model to the pre-trained state for each test task.

## D    ADDITIONAL ANALYSES

For the additional analyses we use pre-trained models that were trained with tasks that have 100 support examples coming from 5 domains. We then deploy them to tasks that have 1) variable number of support domains or 2) variable support set sizes.

Table 4: Analysis of the impact of variable number of domains in the support set of examples – worst 10% tasks test accuracy on CIFAR-C (%).

| Approach | 1 domain | 2 domains | 5 domains | 10 domains | 20 domains |
|---|---|---|---|---|---|
| ERM | $58.3 \pm 0.4$ | $54.2 \pm 0.3$ | $44.3 \pm 0.5$ | $37.9 \pm 0.5$ | $29.4 \pm 0.4$ |
| CML | $57.7 \pm 0.7$ | $54.3 \pm 0.6$ | $44.8 \pm 0.5$ | $39.2 \pm 0.4$ | $30.6 \pm 0.4$ |
| BN | $59.7 \pm 0.5$ | $55.4 \pm 0.5$ | $45.4 \pm 0.7$ | $38.9 \pm 0.7$ | $30.2 \pm 0.6$ |
| CXDA | $62.7 \pm 0.4$ | $58.7 \pm 0.3$ | $49.4 \pm 0.6$ | $43.0 \pm 0.5$ | $33.2 \pm 0.3$ |
| FT-EM | $49.4 \pm 0.6$ | $49.0 \pm 0.5$ | $44.9 \pm 0.6$ | $39.1 \pm 0.5$ | $30.5 \pm 0.4$ |
| FT-IM | $53.1 \pm 0.5$ | $50.7 \pm 0.5$ | $45.6 \pm 0.5$ | $39.6 \pm 0.5$ | $30.7 \pm 0.4$ |

Table 5: Analysis of the impact of variable number of domains in the support set of examples – average test task accuracy on CIFAR-C (%).

| Approach | 1 domain | 2 domains | 5 domains | 10 domains | 20 domains |
|---|---|---|---|---|---|
| ERM | $68.7 \pm 0.3$ | $68.7 \pm 0.3$ | $68.6 \pm 0.3$ | $68.5 \pm 0.2$ | $68.6 \pm 0.2$ |
| CML | $68.8 \pm 0.6$ | $69.2 \pm 0.5$ | $69.5 \pm 0.5$ | $69.4 \pm 0.4$ | $69.5 \pm 0.4$ |
| BN | $69.7 \pm 0.4$ | $69.4 \pm 0.4$ | $69.3 \pm 0.4$ | $69.1 \pm 0.4$ | $69.2 \pm 0.4$ |
| CXDA | $72.2 \pm 0.2$ | $72.1 \pm 0.2$ | $72.0 \pm 0.3$ | $71.9 \pm 0.3$ | $71.9 \pm 0.3$ |
| FT-EM | $69.0 \pm 0.4$ | $69.1 \pm 0.3$ | $69.2 \pm 0.4$ | $69.2 \pm 0.3$ | $69.3 \pm 0.3$ |
| FT-IM | $69.8 \pm 0.3$ | $69.6 \pm 0.3$ | $69.5 \pm 0.3$ | $69.4 \pm 0.3$ | $69.5 \pm 0.3$ |

Finally, we further analyse the adaptation cost of CXDA vs fine-tuning based competitors in terms of wall-clock time. In Figure 4, we compare our CXDA which uses no iterative computation during

Table 6: Analysis of the impact of variable number of examples in the support set – worst 10% tasks test accuracy on CIFAR-C (%).

| Approach | 10 examples | 20 examples | 50 examples | 100 examples | 200 examples | 500 examples |
|---|---|---|---|---|---|---|
| ERM | $0.1 \pm 0.0$ | $21.7 \pm 0.1$ | $38.2 \pm 0.5$ | $44.3 \pm 0.5$ | $48.2 \pm 0.6$ | $50.2 \pm 0.7$ |
| CML | $0.0 \pm 0.0$ | $21.7 \pm 0.2$ | $38.9 \pm 0.5$ | $44.8 \pm 0.5$ | $48.6 \pm 0.5$ | $50.6 \pm 0.4$ |
| BN | $1.2 \pm 1.0$ | $22.8 \pm 0.9$ | $39.5 \pm 0.7$ | $45.4 \pm 0.7$ | $49.4 \pm 0.7$ | $51.3 \pm 0.7$ |
| CXDA | $1.8 \pm 0.9$ | $26.4 \pm 0.7$ | $43.0 \pm 0.5$ | $49.4 \pm 0.6$ | $53.8 \pm 0.6$ | $56.2 \pm 0.8$ |
| FT-EM | $0.8 \pm 0.7$ | $22.4 \pm 0.6$ | $39.0 \pm 0.5$ | $44.9 \pm 0.6$ | $48.9 \pm 0.5$ | $50.8 \pm 0.5$ |
| FT-IM | $1.3 \pm 0.9$ | $22.9 \pm 0.8$ | $39.7 \pm 0.5$ | $45.6 \pm 0.5$ | $49.6 \pm 0.5$ | $51.7 \pm 0.4$ |

Table 7: Analysis of the impact of variable number of examples in the support set – average test task accuracy on CIFAR-C (%).

| Approach | 10 examples | 20 examples | 50 examples | 100 examples | 200 examples | 500 examples |
|---|---|---|---|---|---|---|
| ERM | $68.7 \pm 0.3$ | $68.7 \pm 0.3$ | $68.6 \pm 0.3$ | $68.6 \pm 0.3$ | $68.6 \pm 0.3$ | $68.6 \pm 0.3$ |
| CML | $67.7 \pm 0.3$ | $68.7 \pm 0.4$ | $69.3 \pm 0.4$ | $69.5 \pm 0.5$ | $69.5 \pm 0.5$ | $69.6 \pm 0.5$ |
| BN | $69.3 \pm 0.4$ | $69.3 \pm 0.4$ | $69.2 \pm 0.4$ | $69.3 \pm 0.4$ | $69.2 \pm 0.4$ | $69.3 \pm 0.4$ |
| CXDA | $69.6 \pm 0.3$ | $70.9 \pm 0.3$ | $71.7 \pm 0.3$ | $72.0 \pm 0.3$ | $72.1 \pm 0.2$ | $72.2 \pm 0.3$ |
| FT-EM | $69.1 \pm 0.3$ | $69.2 \pm 0.3$ | $69.1 \pm 0.3$ | $69.2 \pm 0.4$ | $69.2 \pm 0.3$ | $69.2 \pm 0.3$ |
| FT-IM | $69.4 \pm 0.3$ | $69.5 \pm 0.3$ | $69.4 \pm 0.3$ | $69.5 \pm 0.3$ | $69.4 \pm 0.3$ | $69.5 \pm 0.3$ |

adaptation, with the alternatives using 1 and 10 iterations of gradient-descent for adaptation. We can see that CXDA is faster than both versions of fine-tuning even if they use the minimum number of steps available. CXDA also significantly outperforms them in terms of the test score.

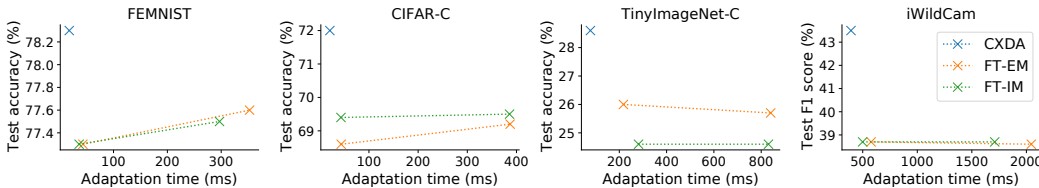

Figure 4: Analysis of test accuracy (%) vs adaptation time (ms) for CXDA and fine-tuning approaches that correspond to TENT (FT-EM) and SHOT (FT-IM) when the baselines use either 1 and 10 steps for fine-tuning.

