# OpenReview forum: "Feed-Forward Latent Domain Adaptation"
_ICLR.cc/2023/Conference — Submitted to ICLR 2023_

### Official Review · Reviewer_J7SR · 2022-10-19

**Confidence:** 3
**Clarity, Quality, Novelty And Reproducibility:** See the above section
**Correctness:** 3
**Technical Novelty And Significance:** 2
**Empirical Novelty And Significance:** Not applicable
**Recommendation:** 3

**Strength And Weaknesses:**

Reasons to accept:

The paper proposes a new research subject, and the results are good.

Reasons to reject:

The paper is extremely verbose, it is full of unnecessary information, it has some missing would-be useful information, and it lacks some visualizations to facilitate the convey of information. All together these have made the article extremely boring to read. As a person who has read many domain adaptation papers, in my opinion this paper is not worth reading, given the amount of information that it offers. Just a few examples (not a full list and not ordered):
- You use too many mini-passages separated by a bold-faced phrase. These confuse the reader, particularly when each one focuses on a specific subject, and occasionally drift from the subject of the section.
- You used these mini-passages in the incorrect sections. For example I don’t expect to see mini-passages called “Latent domain adaptation” or “Feed-forward domain adaptation” in the introduction section, when you already have a full section called “BACKGROUND AND RELATED WORK”. This continues all over your paper. For example, you have a mini-passage called “Preliminaries” in the section called methods, but weren’t you supposed to cover these in the section called background? Even worse, when you read this mini-passage, you would find that it is not actually preliminaries, it is a mix of background and notations.
- You have a mini-passage called “Deployment phase”, aren’t you supposed to cover the model itself, then explain its deployment?
- You have a section called “OBJECTIVE”, weren’t you supposed to explain the model first, and then explain its objective?

   You see, this list goes on and on.

I am not sure about the applicability of the proposed problem setting. This is important because failing to fully justify and evaluate this part, makes me think that the authors have just tried to come up with an unrealistic problem setting to publish a paper for. The authors assume the model cannot use backprop, but cannot we replace the encoder with a smaller one to use a smaller compute? Also based on the examples that you are using in you paper, I am not convinced that why we cannot use source data. You are already using target data, so memory usage shouldn’t be an issue.

In the mini-passage called “Training phase”, you have assumed that the domain of the query image appears in the target set. Why?

**Summary Of The Paper:**

The authors propose a new problem setting: They assume a model is pretrained on source data, then the model is used to predict data in a target domain. They also assume that the resources are low and the method cannot update the parameters—i.e., cannot use backprop.

The proposed model is based on meta-learning. It is pretrained by taking batches as tasks and predicting target labels. The core computation part is to find the target image, in the given target set, that is most similar to the test image using a mechanism similar to self-attention in transformers. Then summing up the embedded vector of this image by the embedded vector of the test image to be used for prediction.

The model is evaluated in four datasets and shows some improvements.

**Summary Of The Review:**

The paper proposes a new problem setting. But the presentation is horrible, and the problem setting needs further experiments to be justified.

---

> ### Author Response · Authors · 2022-11-17
> **Response**
>
> We appreciate the effort and time of the reviewer spent on reviewing our paper. We provide detailed responses to all of the questions.
>
> * **Verbosity and writing style:** We felt it was necessary to explain ourselves in some detail to ensure that all readers understand our new problem setting. We are sorry the reviewer perceived this as verbose and boring and we are very happy to revise more substantially for conciseness if this is the reviewer consensus. We are also happy to revise any particular section for more or less detail if there is a specific request. However, we note that some of the other reviewers have enjoyed reading the paper and found the writing style appropriate, so we believe the paper is written reasonably well.
>
> * **Applicability of the problem setting:** The problem setting is domain adaptation on deployed devices where (i) adaptation needs to happen quickly at low computation cost, and where (ii) the data collected by the mobile device is uncurated and thus cannot be guaranteed to be of consistent relevance to test inputs. This was explained in the first mini-paragraph of the introduction on page 1.
>
>     Far from “proposing an unrealistic setting just to publish a paper”, this project was in fact motivated by our prior practical experience of attempting to deploy domain adaptation to real users in production, and seeing the ways in which existing academic DA algorithms failed under real-world conditions. In particular, we found that they failed because (1) the adaptation sets collected by real users are not as cleanly curated as existing academic domain adaptation algorithms assume, resulting in them failing in practice (we formalise this lack of curation as the latent domain setting), and (2) adaptation must be cheap enough to run on battery-powered devices rather than on a server (which we solve by feed-forward adaptation).
>
> * **Using a smaller model:** Using a smaller model while retaining backprop is unlikely to offer sufficient improvements in speed and latency while maintaining sufficient accuracy. It does not remove the need to iteratively process the data multiple times, and it will decrease prediction quality. Anyway, this comparison is debatable because although our method is backprop-free, we already beat several SotA backprop-based baselines (Table 1, bottom two rows [A,C]) without resorting to a smaller model suggested by the reviewer. If we were to reduce the model size of those baselines it may make them marginally faster, but it will only increase the margin by which we surpass them in terms of accuracy.
> Finally, we remark that as an engineering detail, using a smaller model still doesn’t let you deploy on embedded platforms that simply don’t allow backprop due to only supporting tensorflow lite, etc.
>
> * **Access to source data:** Performing domain adaptation without access to source data is useful to avoid memory and compute overhead, and potentially for privacy reasons. More importantly, source-free domain adaptation is now a well established condition of interest and many papers have been published on this topic [A,B,C]. We should not have to defend studying a condition that already has substantial demonstrated interest from the community.
>
>     [A] Wang etal, Tent: Fully test-time adaptation by entropy minimization. In ICLR, 2021.
>
>     [B] Li et al. Model adaptation: unsupervised domain adaptation without source data. In CVPR, 2020.
>
>     [C] Ling etal, Do we really need to access the source data? Source hypothesis transfer for unsupervised domain adaptation. In ICML, 2020.
>
> * **Domain of the query image in the target set:** First, we emphasise that exactly matching query/test and support/train domain is an assumption made by DA papers prior to ours. In fact our contribution is relaxing this standard assumption. In particular, the relaxed assumption is one where the adaptation/train set contains a mixture of domains, of which at least one matches the domain of the query/test set.
>
> We believe we have been able to address all of the stated weaknesses and questions of the reviewer. We are happy to answer any further questions that you may have, and hope that the reviewer can now recommend acceptance for our paper.

---

> > ### Comment · Reviewer_J7SR · 2022-12-11
> > **Re: Response**
> >
> > None of my comments have been addressed.
> > The authors believe that the presentation of their paper is fine, the problem setting does not need justification, the experiments are sufficient, and also believe that their assumption about the sampling of batches are being adopted by other DA papers.
> >
> > I hold my verdict: I gave concrete examples that the paper mixes explanations and follows a confusing order. I stress that despite the authors arguing that there are cases that we cannot access to source data and cannot use backprop, their experiments are in regular setting where there is no reason that we limit ourselves to pretrained source models. Their argument about sampling is also factually incorrect.
> >
> > Final words: Just a reminder that it is the reviewers task to state whether the authors have addressed their comments or not, and the AC’s task to make the judgment.

---

> > > ### Author Response · Authors · 2022-12-12
> > > **Follow-up Response**
> > >
> > > Thanks for getting back to our response. We follow-up on the claims in the reviewer’s response, explaining further misunderstandings and also addressing the other comments. We hope that the AC will recognize the reviewer’s comments were addressed appropriately.
> > >
> > > **Presentation of the paper:** It is not only our belief that the paper is fine (well-written), also other reviewers mention this in their reviews. Hence we believe the writing style is acceptable - noting that a large part of this review only focuses on the writing style. Thanks for giving us the concrete examples regarding presentation; we hoped that by giving our reasoning for why we wrote the paper in the way we did, the reviewer would agree the current style is reasonable - especially because other reviewers find the paper well-written.
> > >
> > > **Justification of the problem setting:** We provide justification within the first mini-paragraph of the introduction on page 1 and also in *Applicability of the problem setting* section of our response. Even though we clearly provide the justification, the reviewer states “the authors believe that the problem setting does not need justification”.
> > >
> > > **Experiments:** The reviewer did not mention specific experiments, so it is hard to see why the reviewer indicates that the experiments would be insufficient. In case this refers to the idea of using a smaller model and adapting it using backpropagation, we have explained why this strategy is not meaningful in our case.
> > >
> > > **Sampling of batches adopted by other DA papers:** Let us rephrase this as there was likely a misunderstanding of our response. We meant that in general there is only one target domain and the actual test examples also come from this target domain (hence the domains are matched). We referred to this domain as the support/train domain in the context of how our adaptation tasks are constructed - test (query) examples also come from this domain.
> > >
> > > **Experimental setting**: The fact that the datasets themselves can also be used for other settings does not mean that we cannot repurpose the datasets for the specific setting that we study. Our experiments are NOT in the regular setting (regular = not using pre-trained source-domain models) because of the design of our experiments and not because of the datasets that we use (e.g. when adapting to a new test task, we do not give access to the source data used for pre-training).
> > >
> > > We operate within the source-free domain adaptation paradigm, which is a restriction that we bring in. The paradigm of source-free domain adaptation has been widely studied before and has substantial practical benefits [A-G], so we believe it is reasonable to use it as a restriction for how the adaptation and evaluation is done. Further, note that we already compare with standard backprop-based alternatives too, and we beat them.
> > >
> > > **Argument about sampling:** We have explained this misunderstanding in the point about *Sampling of batches adopted by other DA papers*.
> > >
> > > References:
> > >
> > > [A] Yang et al., Attracting and Dispersing: A Simple Approach for Source-free Domain Adaptation. NeurIPS’22
> > >
> > > [B] Zhang et al., Divide and Contrast: Source-free Domain Adaptation via Adaptive Contrastive Learning. NeurIPS’22
> > >
> > > [C] Jing et al., Variational Model Perturbation for Source-Free Domain Adaptation. NeurIPS’22
> > >
> > > [D] Lee et al., Confidence Score for Source-Free Unsupervised Domain Adaptation. ICML’22
> > >
> > > [E] Kundu et al., Balancing Discriminability and Transferability for Source-Free Domain Adaptation. ICML’22
> > >
> > > [F] Kundu et al., Universal Source-Free Domain Adaptation. CVPR’20
> > >
> > > [G] Liang et al., Do We Really Need to Access the Source Data? Source Hypothesis Transfer for Unsupervised Domain Adaptation. ICML’20

---

> > > > ### Comment · Reviewer_J7SR · 2022-12-13
> > > > **Re: Follow-up Response**
> > > >
> > > > the authors mostly reiterated their previous comments.
> > > >
> > > > ps. despite having dozens of logistical issues to point out--including the authors attitude to reviewing, to writing a response, to communicating w/ reviewers--i would rather to remain professional. i have spent enough time on working on this paper.

---

> > > > > ### Author Response · Authors · 2022-12-13
> > > > > **Response**
> > > > >
> > > > > Apologies if any parts of our responses appeared as too assertive to you, we tried to remain as professional as possible in our responses - and we are grateful for the time and effort you have spent on writing the review.

---

### Official Review · Reviewer_zikv · 2022-10-23

**Confidence:** 4
**Correctness:** 2
**Technical Novelty And Significance:** 2
**Empirical Novelty And Significance:** 1
**Recommendation:** 3

**Clarity, Quality, Novelty And Reproducibility:**

There are some typos in the manuscript. The authors didn’t clearly demonstrate the strength of the feed-forward training. The technical novelty is very limited.

**Strength And Weaknesses:**

Strength:
1. The authors proposed a new setting for source-free domain adaptation. This setting seems more practical in real-world scenarios compared with the conventional SFDA.

2. The writing of this manuscript is clear and easy to follow. The overall presentation is nice, including smooth sentences, nice figures, and organised paragraphs.

3. The pipeline of the method is technically sound and provides detailed implementation of the proposed method.

Weakness:

1. In Section 3.1, the authors mentioned that “While the conventional domain adaptation setting assumes that xq and xs are all drawn from a common distribution…”. In the conventional domain adaptation, however, they usually belong to different domains.

2. Compared with ARM, the proposed method seems a bit incremental, it is just the simple combination of ARM and cross-attention module. The authors didn’t provide strong motivation for combining these techniques. In addition, neither ARM nor cross-attention module was proposed by the authors.

3. Typo in Section 3.1: “To train a model than can be used as described above”. “than” should be “that”.

4. In Sect. 3.3, the author mentioned that “image-to-image attention is more suitable for domain adaptation than patch-based option because the overall representation should better capture the nature of the domain rather than a patch.” Is there any theory that supports this conclusion?

5. In my opinion, it is necessary to compare the proposed method with TENT and SHOT even though they are based on back-propagation. Although the proposed method does not require back-prop, this is a detail of the specific implementation and is not relevant to the task.

6. Even with the feed-forward, the proposed method still requires several iterations for training. So, what are the advantages of using feed-forward?

7. As the training time is heavily affected by the number of iterations of the method, the numbers in Table 3 are not convincing enough. Which method performs best with the same training time? Which method requires less training time to achieve the same performance? Furthermore, comparisons with TENT and SHOT are still missing from Table 3.




**Summary Of The Paper:**

In this paper, the authors proposed the Latent Domain Adaptation setting where the a source model should be adapted to a target dataset that contains a mixture of unlabelled domain-relevant and domain-irrelevant examples. The “latent'' means that we are not given the domain label in the training phase. To handle this problem, the authors proposed a feed-forward method which incorporates the cross-attention module with an episodic meta-learning framework. The authors have carried out extensive experiments to verify the performance of the proposed method. Furthermore, the authors analysed the training and inference time to show the strength of the proposed feed-forward method.

**Summary Of The Review:**

Considering the limited technical novelty and insufficient comparison, e.g., with TENT and SHOT, the reviewer would not be able to recommend an acceptance.

---

> ### Author Response · Authors · 2022-11-17
> **Response (1/2)**
>
> Thank you very much for the review and for the effort and time spent reviewing our paper. We are especially grateful that you highlight the practicality of our proposed setting, clear writing and also the soundness of the method. We provide answers to the questions from the review:
>
> * **Domain adaptation:** Sorry for the confusion. $x_q$ and $x_s$ do not refer to target and source domain as the reviewer has interpreted. They refer to the test set and unlabeled training set of the target domain respectively. Conventional UDA assumes these target train and test samples are drawn IID from the target domain. Our latent domain setting assumes that they are drawn from a target set that is comprised of a mixture of latent domains.
>
>     We have updated the paper to improve the clarity of this part. In particular for the source-free domain adaptation setting, we have unlabelled “training” examples from the target domain that are used for adaptation (equivalent to our support examples) and then test examples on which we evaluate the quality of adaptation (equivalent to our query examples). These typically come from the same domain in existing set-ups.
>
> * **Method novelty and motivation:** The focus of our paper is on introducing the new highly practical problem setting and addressing it in a suitable way, providing the first solution to it. We believe it is not to the detriment of our paper that this solution is based on a smart combination of meta-learning and a cross-attention mechanism.
>
>     The motivation for combining these techniques is to solve our new latent domain problem setting. As explained in the schematic in Figure 1, this corresponds to adaptation where the target train set consists of a mixture of latent domains, which are of unknown relevance to the test image. How can we know which images within the adaptation set are relevant to the test image, (and thus avoid negative transfer by inappropriately adapting based on irrelevant images)? This is exactly the contribution of our innovative use of cross-attention.
>
> * **Typo:** Thanks for mentioning it, we have fixed it in the revised version.
>
> * **Why image-to-image attention?** First, please note that our image-to-image attention operates on a vectorised list of patches, so it is effectively patch-to-patch but where only corresponding patches ImageA(i,j) to ImageB(i,j) are compared across images. The difference with patch-patch attention is that non-corresponding patches (i,j) to (k,l) would be compared.
>
>     To answer the actual question, we could use patch-to-patch attention and it might work in principle, but we chose not to because: (1) it would be substantially more expensive (cost rather than accuracy) and (2) it might not bring much benefit because since it’s a more complex architecture that would be be easier to overfit, (3) we expect most of the benefit to come from image-to-image attention. It’s not intuitive that there is much more information to gain from comparing non-corresponding patches across images. We now include an ablation comparing image-to-image attention and patch-to-patch attention (CXDA P2P). The results show that P2P usually performs slightly worse despite the greater computational cost, which we attribute to greater ease of overfitting this more complex model.
>
>
>     Main results on synthetic and real-world benchmarks: average and worst-case (worst 10% tasks) test performance, with standard error of the mean across 3 random seeds. Accuracy is reported for all except iWildCam, where F1 score is used (%).
>
> | | FEMNIST | | CIFAR-C | | TinyImageNet-C | | iWildCam | |
> | --- |--- |--- |--- |--- |--- |--- |--- | --- |
> | Approach | W10\% | Avg | W10\% | Avg | W10\% | Avg | W10\% | Avg |
> | ERM | 52.7 $\pm$ 1.4 | 77.2 $\pm$ 0.9 | 44.3 $\pm$ 0.5 | 68.6 $\pm$ 0.3 | 4.8 $\pm$ 0.2  | 26.4 $\pm$ 0.4 | 0.0 $\pm$ 0.0 | 38.7 $\pm$ 0.8 |
> | CML | 50.4 $\pm$ 1.3 | 76.0 $\pm$ 0.9 | 44.8 $\pm$ 0.5 | 69.5 $\pm$ 0.5 | 4.8 $\pm$ 0.5 | 25.7 $\pm$ 0.6 | 0.0 $\pm$ 0.0 | 38.7 $\pm$ 1.1 |
> | BN | 52.2 $\pm$ 1.5 | 78.0 $\pm$ 0.7 | 45.4 $\pm$ 0.7 | 69.3 $\pm$ 0.4 | 5.9 $\pm$ 0.2 | 27.7 $\pm$ 0.3 | 1.9 $\pm$ 1.1 | 42.5 $\pm$ 0.8 |
> | CXDA | 53.3 $\pm$ 0.6 | 78.3 $\pm$ 0.0 | 49.4 $\pm$ 0.6  | 72.0 $\pm$ 0.3 | 6.5 $\pm$ 0.2 | 28.6 $\pm$ 0.3 | 3.6 $\pm$ 1.5 | 43.5 $\pm$ 1.5 |
> | CXDA P2P | 52.0 $\pm$ 0.7 | 77.3 $\pm$ 0.3 | 45.6 $\pm$ 0.3 | 69.8 $\pm$ 0.2 | 6.8 $\pm$ 0.2 | 28.9 $\pm$ 0.1 | 4.1 $\pm$ 0.9 | 42.9 $\pm$ 1.3 |

---

> ### Author Response · Authors · 2022-11-17
> **Response (2/2)**
>
> * **TENT and SHOT:** We do already provide this comparison and the outcome favours CXDA. Please note that fine-tuning with entropy minimisation and infomax (already included in our paper and denoted as FT-EM and FT-IM respectively in Table 1 and 3) already correspond to applications of TENT and SHOT algorithms to our problem setting.
>
>     We remark that these results are presented as context rather than direct comparison because they use backprop. This is not a minor algorithm-specific implementation detail as it  means that they are around 10x slower for adaptation (see Table 3, left), which is salient for our motivating application of near real-time domain adaptation on a mobile device. They are also not designed for latent domain application, which explains their comparatively weak performance despite the massive advantage they have by using backprop.
>
> * **Iterations:** The iterations are only used during meta-training (pre-training) and this is completely separate from the adaptation that happens on new tasks when deployed to users. During adaptation there are no iterations and being feedforward means that we can do the adaptation extremely quickly, compared to methods that use iterations for adapting to the given task.
>
> * **Training time:** We would like to clarify that there are two relevant computation costs: (i) time required for source model (meta)training, and the (ii) actual domain adaptation time. The relevant cost is the latter domain adaptation time, since this is what the user sees, and because the prior time-spent on meta-learning/source model training is amortised across tasks.
>
>     Note that this latter cost evaluation is both standard practice in amortised learning studies  [A - see Fig 7 therein]), and in source-free domain adaptation. In source-free DA, we mainly evaluate the adaptation time to a new dataset: we do not consider the source model training time since this is not relevant to the actual adaptation, and can be amortised across target datasets. Our meta-learning process is part of the source model training time, and thus not something that would be measured when measuring the adaptation cost of source free-DA or amortised learning.
>
>     With this in mind, we can observe that the time to adapt to a new domain is around 10x faster for our CXDA than standard backprop approaches (Table 3, left - recall that FT-EM and FT-IM correspond to TENT and SHOT), and similar to other non-backprop approaches. This 10x speedup answers the reviewers question “what is the advantage of using feed-forward?”.
>
>     Although this is not a standard or fair comparison, we further note that our *total* time, including (i) source model training + (ii) adaptation cost, is similar or faster than the time required for backprop-based competitors (Table 3, right).
>
>     To specifically answer the reviewer’s comment: “As the training time is heavily affected by the number of iterations of the method, the numbers in Table 3 are not convincing enough. Which method performs best with the same training time? Which method requires less training time to achieve the same performance?”
>     1) The relevant time is the domain adaptation time for the target domain [A], not the source model (meta)-training time. The domain adaptation time of CXDA independent of the number of iterations for source model training, and CXDA uses no iterative computation during adaptation to the target domain.
>     2) By comparing Table 1 and Table 3, one can conclude that (i) CXDA performs better than TENT and SHOT with less adaptation time (Table 3, left), (ii) Even if including both source model time and adaptation time together, CXDA performs better than TENT and SHOT with lesser total time for all datasets besides the smallest FEMNIST (Table 3, right).
>     3) To more clearly convey these results in a way that directly answers the reviewer’s question, we have added a new plot  to Appendix D (Figure 4) that shows performance vs adaptation time for CXDA vs TENT and SHOT using different numbers of iterations. We can see that (i) CXDA is best, with greater accuracy for equal or less training time, and (ii) CXDA needs less training time to achieve good performance.
>
>     [A] Requeima et al., Fast and Flexible Multi-Task Classification Using Conditional Neural Adaptive Processes. NeurIPS’19
>
> We believe we have been able to address all of the stated weaknesses and questions of the reviewer and hope that the reviewer can recommend acceptance for our paper.

---

### Official Review · Reviewer_tuYo · 2022-10-24

**Confidence:** 4
**Clarity, Quality, Novelty And Reproducibility:** See the weakness.
**Correctness:** 3
**Technical Novelty And Significance:** 2
**Empirical Novelty And Significance:** 2
**Recommendation:** 3

**Strength And Weaknesses:**

Strength:
the writing of this papers seems easy to follow and it demonstrates its method from several perspectives.

Weakness:
1.	There are several implementation details associated with the proposed method such as the augmentation, cross attention, meta learning. It would be better to have  the ablation studies on decomposing the each components.
2.	I am concerned the performance boost with the relatively large computational time. Although in table 1, the authors shows the improvements but coming with larger computations.
3.	The author claims privacy in the paper. However, there are not many privacy baselines are used as the baselines. I would recommend the authors to compare with some privacy related domain adaption such as black box related domain adaption.
4.	I also have the questions regarding the scalability of the proposed method. With the meta-training, how does the proposed method formulate on the larger dataset or settings? And what are there performance? It would be better to extend the proposed method on the larger dataset and with the detailed running time. I suspect that the running time will be much larger compared to the baselines.


**Summary Of The Paper:**

This paper proposes to meta-learn a network capable of embedding the mixed-relevance target dataset and dynamically adapting inference for target examples using cross-attention. It further analysis the method performance in the domain-supervised adaption.

**Summary Of The Review:**

----------------------------Updates---------------------------------
After reading the response and other reviewers' comments, I would like to keep my score.

---

> ### Author Response · Authors · 2022-11-17
> **Response**
>
> We appreciate the effort and time of the reviewer spent on reviewing our paper. We provide detailed responses to all of the questions.
>
> * **Implementation details:** Meta-learning is inherently associated with a choice of meta-parameters, which in our case are represented by the cross-attention module. It is not possible to do meta-learning without meta-parameters, or use the meta-parameters without meta-learning, so we believe the request to decompose these two is not applicable. Data augmentation is a standard tool, and we use it consistently across all of the approaches that we compare, so we are very surprised that the reviewer sees it as a component of our method.
>
> * **Computational time:** We are surprised that the reviewer sees computation cost as a drawback, as our method is faster than the conventional gradient-based competitors during adaptation as shown comprehensively in Table 3 (left side). In more detail, we would like to clarify that there are two relevant computation costs: (i) time required for source model (meta)training, and the (ii) actual domain adaptation time. The relevant cost is the latter domain adaptation time, since this is what the user sees, and because the prior time-spent on meta-learning/source model training is amortised across tasks.
>
>     Note that this latter cost evaluation is both standard practice in amortised learning studies  [A - see Figure 7 therein], and in source-free domain adaptation. In source-free DA, we mainly evaluate the time required to adapt to a new dataset: we do not consider the source model training time since this is not relevant to the actual adaptation, and can be amortised across target datasets. Our meta-learning process is part of the source model training time, and thus not something that would be measured when measuring the adaptation cost of source free-DA or amortised learning.
>
>     With this in mind, we can observe that the time to adapt to a new domain is around 10x faster for our CXDA than standard backprop approaches (Table 3, left), and similar to other non-backprop approaches. Finally, although this is not a standard comparison, we further note that our *total* time, including (i) source model training + (ii) adaptation cost, is similar or faster than the time required for backprop-based competitors (Table 3, right). Therefore overall, it is an inaccurate reading of our results to say that our accuracy improvements come at increased computational cost, and in fact the opposite is true.
>
>     [A] Requeima et al., Fast and Flexible Multi-Task Classification Using Conditional Neural Adaptive Processes. NeurIPS’19
>
> * **Privacy:** To clarify, we do not make any claim about contributing to privacy. We just mentioned this as one of several reasons to motivate our focus on the source-data free condition, where source data is not revisited during adaptation. The source-free condition is also separately necessitated by the low-latency adaptation requirement of our motivating application.
>
>     In terms of comparisons, we emphasise that *all the baselines* that we already compare provide an equal degree of privacy in that none of them require access to the source data. So we already provide the relevant and fair comparisons.
>
> * **Scalability:** Our source model (meta) training process samples tasks from the dataset in a mini-batch manner. So the cost is the standard (minibatch cost) x (number of iterations). It is linear in the dataset size assuming a fixed number of epochs, as per many other common learning algorithms. Thus the source dataset can be of any size and there are no restrictions. In fact, we already use large datasets in our experiments - the iWildCam dataset is a large dataset that is >10GB, has >200k examples and >300 domains. The performance is strong and the runtime is very similar to that of the baselines (Table 3), contrary to the expectations of the reviewer. The method can also scale also to a larger number of support examples and domains per task, as already shown in Appendix D.
>
> We believe the above explanations clearly address all the claimed weaknesses and questions of the reviewer and hope that the reviewer can recommend acceptance for our paper. We are happy to answer any further questions the reviewer may have.

---

### Official Review · Reviewer_wyNm · 2022-10-24

**Confidence:** 2
**Clarity, Quality, Novelty And Reproducibility:** 1) The paper explores a new setting w…
**Correctness:** 3
**Technical Novelty And Significance:** 3
**Empirical Novelty And Significance:** 3
**Recommendation:** 6

**Strength And Weaknesses:**

1) This paper explore new setting about latent domain adaptation: source-free and forward-pass, which is a new highly practical setting.

2) Authors try to give clear explanation about this setting, and show why it is interesting in practical application.

3) The paper is well-written and easy to follow.

4) The experiment is sufficient to support the proposed method.


**Summary Of The Paper:**

In this paper, authors focus on source free latent domain adaptation. Here the content or label of  target data overlaps with the one of the source data. And authors doest not get access to the source data when using target data. Furthermore, authors also strict setting where the network is not updated which is backward-free. To achieve this goal, authors explore the probability of using meta-learning, and propose to use cross-attention to select relevant examples for adaptation. The experiment results demonstrate the effectiveness of the proposed method.





**Summary Of The Review:**

This paper use the meta-learning method to explore source-free latent domain adaptation, which sounds interesting. However, for forward-pass, it is less convincing.

---

> ### Author Response · Authors · 2022-11-17
> **Response**
>
> Thank you very much for the encouraging review and for the effort and time spent reviewing our paper. We are especially grateful that you highlight that our proposed setting is highly practical and also that the paper is well-written. We provide answers to the questions from the review:
>
> * **Meta-learning:** We agree we use meta-learning in a fairly standard way, but it is a tool that enables us to solve the new highly challenging problem setting, so we do not see it as a weakness.
>
> * **Size of the empirical benefit:** Our CXDA provides consistent benefit. Especially for the harder datasets CIFAR-C, TinyImageNet-C and iWildCam, we achieve a clear margin of (2.5%, 4.0% 4.8%) average case and (3.8%, 1.7%, 3.6%) worst case over the standard baseline SHOT (FT-IM), despite that the latter uses much slower backprop and we do not.
>
> * **Significance of forward pass-based adaptation:** We envisage that our method would be used to perform adaptation to new tasks quickly on the fly, where efficiency and latency of adaptation to a new data distribution is crucial (e.g., on a mobile robot entering a new environment). In this case standard backpropagation-based methods would not be practically viable, while our CXDA has no problem. The runtime difference between CXDA and backprop-based approaches is an order of magnitude, as shown in Table 3 (left side).

---

### Official Review · Reviewer_tHMU · 2022-10-24

**Confidence:** 4
**Correctness:** 4
**Technical Novelty And Significance:** 4
**Empirical Novelty And Significance:** 3
**Recommendation:** 8

**Clarity, Quality, Novelty And Reproducibility:**

<Clarity>

This paper is well-written and easy to follow. I enjoyed reading it.
Minor concerns are as follows:
- it might be better to change the notation of the classifier in Eq. (1) from f to any other letter.
- z represents a domain in Eq. (1) but extracted features at the end of Section 3.3. It is also encouraged to change either notation.

<Quality>

As far as I understand, this submission is technically sound.

<Novelty>

Although the proposed method follows a similar flow to ARM, there are two critical differences as steted in Section 3.1. Especially, the second one, which is robustness against domain-irrelevant samples in the target dataset, is practically important and is effectively tackled by utilizing a cross-attention mechanism, which provides sufficient originality to this study.

<Reproducibility>

The paper provides sufficient implementation details in Section 4.3 and the appendix.


**Strength And Weaknesses:**

<Strength>

- Since the proposed method does not require any training process with back-propagation operations, it should be easy to use and can be applied to a wide range of applications.
- The proposed method is also useful in terms of robustness against domain-irrelevant samples in the target dataset.
- The proposed method consistently achieves good performance in several benchmark datasets.
- This paper is well-written and easy to follow. I enjoyed reading it.

<Weakness>

- I do not have any major concern.
- Shown below are minor things (almost just out of curiosity).
	- Do we need to fix the size of the support set? As a mechanism, the cross-attention module can process a variable size of the support set. Allowing this might be useful, because it makes it possible to handle a various size of a target dataset as is.
	- Can we use the proposed method in an online manner like test-time adaptation? Since LN(f(x_s)) can be stored for the subsequent inference process once we compute it, we can conduct online adaptation by using the proposed method, if we have a strategy to properly update the stored feature with LN(f(x_q)). It would be great if we can use the proposed method as an unified method to solve both source-free domain adaptation and test-time adaptation. (I understand that the above makes the proposed method much similar to ARM, though)




**Summary Of The Paper:**

This paper presents a new source-free domain adaptation method that can work without back-propagation and is capable of handling a target dataset containing both domain-relevant and domain-irrelevant samples (called LDA setting). The proposed method firstly embeds the target dataset into features and conduct adaptive inference for target samples using cross-attention mechanism with the embedded features. The modules for embedding, cross-attention, and classification are jointly trained via a scheme of meta-learning. Experimental results with several popular datasets show that the proposed method works well especially in the LDA setting.

**Summary Of The Review:**

The proposed source-free domain adaptation is sufficiently novel and empirically works well. The manuscript is well-written. I vote for "accept."

---

> ### Author Response · Authors · 2022-11-17
> **Response**
>
> Thank you very much for the encouraging review and for the effort and time spent reviewing our paper. We are especially grateful that you highlight the usefulness of our method, its robustness, good results and also the quality of writing. We provide answers to the questions from the review:
>
> * **Fixed size of the support set:** The support set does not need a fixed size. The cross-attention module can process a variable sized support set.
>
> * **Online manner:** Yes, our method is designed to support online adaptation. Compared to existing test-time adaptation methods, our method can work even with a single query example rather than requiring many of them - support for the streaming mode is a major benefit of our method.
>
> * **Notation:** Thanks for the suggestions, we have improved the notation in the revised version. We now refer to the domain as $d$. Note that the classifier is parametrized by parameters $\phi$ and $f$ is a function of the parameters that we pass to it.

---

### Author Response · Authors · 2022-11-17
**Shared Message to AC and All Reviewers**

Dear AC and reviewers,

We sincerely thank you for the time and efforts spent for reviewing our paper. We appreciate that reviewers have found our paper as “highly practical”, “well written and easy to follow” and that it  “can be applied to a wide range of applications”. We believe the main issues flagged in some of the reviews were points of clarity and misunderstandings. We believe we have comprehensively addressed these.

To summarise the most substantial points of confusion and their resolution:
* **Comparison to backprop-baselines such as TENT and SHOT.** Reviewer zikv thinks this comparison is missing, but it is in fact there in Table 1, and shows that CXDA beats these alternatives.
* **Computational time.** Reviewer tuYo and zikv were concerned about this. We clarified that there are two relevant quantities (i) source domain model training time and (ii) time required to adapt the source model to a new domain. Prior papers have focused on (ii) and so do we. By this measure of adaptation time we are around 10x faster than competitors (Table 3, left) while having better accuracy (Table 1). For completeness, we also added up time (i) and (ii) for each competitor, and our total source model training + adaptation time is still similar or better than competitors  (Table 3, right), while having better accuracy (Table 1).
* **Problem setting.** Reviewer J7SR doubts the real-world significance of our problem setting (disagreeing with the other reviewers in this regard). We clarified that our study was motivated by our prior experience attempting to deploy SotA domain adaptation algorithms to real users in production, finding that (i) datasets collected by real users are not as well curated as academic datasets, which broke existing algorithms and motivated our latent domain setting. (ii) Adaptation may be required to run on low-capability battery powered devices, which motivated our focus on feed-forward algorithms. Nevertheless, our experiments show our feed-forward algorithm can beat SotA backprop-based alternatives while being substantially faster.

---

### Author Response · Authors · 2022-12-06
**Gentle Reminder**

Dear reviewers,

We appreciate the time and efforts you have spent reviewing our paper. We have addressed all questions and weaknesses in our earlier replies, and we hope you can get back to us to let us know whether you agree your concerns have been resolved. If you have any further concerns or questions, please let us know too.

We appreciate that reviewers have found our paper as “highly practical”, “well written and easy to follow” and that it “can be applied to a wide range of applications”. We believe the main issues flagged in some of the reviews were points of clarity and misunderstandings, and we hope that also the remaining reviewers can now recommend acceptance.

Thank you very much!

Authors

---

### Decision · Program_Chairs · 2023-01-20

**Decision:**

Reject

**Justification For Why Not Higher Score:**

The learning setup was not sufficiently motivated, and some simple and natural baselines are missing. The empirical results are not strong, and scalability may be an issue.

**Justification For Why Not Lower Score:**

N/A

**Metareview: Summary, Strengths And Weaknesses:**

In the context of episodic meta-learning, this paper introduces a new domain adaptation (DA) setup where domains are unknown ("latent") and test samples are not grouped to any explicit domain. This is motivated by their work with real DA, where test samples were not cleanly tagged into the new domains.  They propose a meta-learning approach that uses cross attention between test instances and adaptation sets.  As a second novelty, they look into an approach that does not involve back-prop optimization at adaptation time, to reduce computation at inference. The approach is tested on four benchmarks, yielding small improvement that is statistically significant in CIFAR-C and tiny imagenet.

Reviewers had split views about this paper, three reviewers recommended clear rejection and two recommended acceptance.The main concerns were around the empirical evaluations and scalability, and also around technical novelty. Authors submitted a rebuttal, but all reviewers decided to keep their scores.

Two reviewers were concerned that training time may be prohibitively large due to the meta-learning procedure. Author claimed that training time is not important, because it is amortized across tasks. They also point out that they experiment with a large dataset iWildCam is >10GB. Authors state: "we do not consider the source model training time since this is not relevant to the actual adaptation". The point is that meta-learning approaches spend much more time at training time to yield models that can be adapted better or more quickly. They therefore tradeoff test-time adaptation with training time.

One reviewer was concerned with the quality of presentation. This AC does not see any unusual issues with that aspect.

Regarding the new setup of latent DA. In their motivating examples, it seems natural to group samples into domains based on metadata (location, time of acquisition), or by inferring a latent domain (clustering or manifold learning). Even their Figure 1 makes it clear that it would be easy to assign samples to domains (camera 1, night). For a new learning setup, one would expect to see examples where it is not possible to group samples into domains. In this context, iWildCam does not seem like the best datasets for this task, because the "domain" (camera ID) is not necessarily correlated with visual properties of the sample (as the authors point out). A very natural and important baselines to compare with, would be to infer a domain at test time. In a way, CXDA does exactly that in an implicit way at test time, and I would expect to see comparisons with approaches that infer the domain explicitly.

A potential benefit of the method is in terms of robustness to samples with misclassified domains. Perhaps CXDA yields stronger improvement over standard DA under such domain label noise?

Based on these technical concerns, the paper is not ready for publication in ICLR.





**Summary Of Ac-Reviewer Meeting:**

N/A